# Selection Behavior of the Beet Armyworm, *Spodoptera exigua* (Hübner) Between Bt Maize and Conventional Maize Plants

**DOI:** 10.3390/insects16101059

**Published:** 2025-10-17

**Authors:** Cheng Song, Xianming Yang, Guodong Kang, Limei He, Wenhui Wang, Xiang Han, Yujiao Xie, Kongming Wu

**Affiliations:** 1State Key Laboratory for Biology of Plant Diseases and Insect Pests, Institute of Plant Protection, Chinese Academy of Agricultural Sciences, Beijing 100193, China; songcheng990624@163.com (C.S.); yangxianming@caas.cn (X.Y.); kanggd95@163.com (G.K.); w975480209@163.com (W.W.); 19988158824@163.com (X.H.); xieyujiao02@163.com (Y.X.); 2Institute of Urban Agriculture, Chinese Academy of Agricultural Sciences, Chengdu 610299, China; helimei91@163.com; 3Institute of Insect Sciences, College of Agriculture and Biotechnology, Zhejiang University, Hangzhou 310058, China; 4State Key Laboratory of Agricultural and Forestry Biosecurity, Fujian Agriculture and Forestry University, Fuzhou 350002, China

**Keywords:** Bt maize, *Spodoptera exigua*, feeding behavior, oviposition behavior, dispersal behavior, resistance management

## Abstract

**Simple Summary:**

Bt crops are a key strategy for controlling *Spodoptera exigua* (Hübner), yet the behavioral characteristics of this pest on Bt and non-Bt plants remain unexplored. This study demonstrates that *S. exigua* larvae prefer to feed on non-Bt maize and exhibit antifeedant and avoidance behavior toward Bt maize expressing Cry1Ab + Vip3Aa19 proteins. Although female moths showed no oviposition preference between Bt and non-Bt maize plants under undamaged conditions, they preferentially oviposited on Bt maize when non-Bt maize plants were damaged. Under the seed-mixture refuge pattern, *S. exigua* larvae exhibited frequent interplant movement between Bt and non-Bt maize plants. Increasing the proportion of non-Bt maize significantly enhanced larval dispersal distances and raised the risk of transit damage to Bt maize plants. These findings clarify the behavioral responses of *S. exigua* to Bt and non-Bt maize plants, and provide scientific evidence for optimizing refuge strategy to delay resistance evolution.

**Abstract:**

Establishing refuges is a primary strategy for managing resistance in target pests against Bt maize. The larval feeding and dispersal, and adult oviposition behaviors of *Spodoptera exigua* (Hübner) on Bt and non-Bt maize plants are critical factors in determining optimal refuge configurations. This study employed laboratory and field experiments to evaluate the larval feeding and dispersal behaviors, as well as the oviposition preferences of *S. exigua* moths, on Bt (Cry1Ab + Vip3Aa19) and non-Bt maize plants. Results showed that as time of the choice test increased, the larval selection rate on Bt maize leaves declined progressively, with all instars (1st–5th) preferring to feed on non-Bt maize. After 48 h, the selection rates of larvae for non-Bt and Bt maize were 40.63–66.25% and 9.38–33.75%, respectively. Female moths exhibited no significant oviposition preference between Bt and non-Bt plants under undamaged conditions; however, when non-Bt maize was infested by the larvae, females preferentially oviposited on Bt maize plants (73.55%). Under the seed-mixture refuge pattern in field conditions, increasing the proportion of non-Bt maize significantly enhanced larval dispersal distances and facilitated larval transit damage between Bt and non-Bt plants. Our research clarifies the behavioral patterns of *S. exigua* on Bt and non-Bt maize, provides a scientific basis for optimizing refuge strategy to delay the development of resistance.

## 1. Introduction

The beet armyworm, *Spodoptera exigua* (Hübner) (Lepidoptera: Noctuidae), is a southern Asian native and a major agricultural pest that breeds year-round in subtropical regions [1,2]. Its larvae are highly polyphagous, infesting various staple and economically important crops such as maize, cotton, soybeans, cabbage, and scallions [3]. Adults possess high fecundity and strong long-distance migratory capacity, with field monitoring recording dispersal distances up to 3500 km [4]. These biological traits greatly facilitate population outbreaks and geographic expansion, with severe infestations reported across Asia, Australia, the Americas, Africa, and Europe [3,5]. In China’s Huang-Huai-Hai summer maize region, *S. exigua* has recently caused significant damage during the vegetative growth stage, especially in maize-soybean strip intercropping systems, where larval densities can reach up to 80 individuals per 100 plants [6,7]. Current control measures rely heavily on chemical insecticides; however, the pest’s tendency to conceal itself under maize foliage or in the soil near the stem base during the daytime reduces control efficacy. Concurrently, resistance to several commonly used insecticides, such as beta-cypermethrin, chlorantraniliprole, indoxacarb, and chlorpyrifos, has developed, posing risks to environmental safety and food security [8,9]. Therefore, developing efficient and environmentally friendly control technologies is urgently required in maize production.

Bt (*Bacillus thuringiensis*) insect-resistant maize is a crucial component of integrated pest management (IPM) strategies targeting lepidopteran and coleopteran pests [10]. Since its commercial deployment, Bt maize has reduced insecticide use globally, improved crop yield and quality, and effectively controlled major pests including *Ostrinia nubilalis* (Hübner) and *Spodoptera frugiperda* (J. E. Smith), generating significant economic, social, and ecological benefits [11,12,13]. Several Bt events expressing Cry1Ab/Cry2Aj, Cry1Ab/Vip3Da, and Cry1Ab + Vip3Aa19 have been evaluated for their efficacy against *S. exigua* [14,15,16]. Among these, Bt (Cry1Ab + Vip3Aa19) maize (event DBN3601T) demonstrated over 95% control efficacy and significantly reduced plant damage [16]. This event has been industrially deployed in China.

Accumulated global experience indicates that the evolution of pest resistance is a critical factor affecting the sustainable use of Bt crops [17]. The number of confirmed cases of field-evolved resistance to Bt crops increased from 3 in 2005 to 26 in 2020, involving eight maize pests (six lepidopteran and two coleopteran species) [18]. Resistance threatens the efficacy of eight widely deployed Cry toxins, including Cry1Ab, Cry2Ab, Cry1Fa, and Cry1A.105, in five countries [18]. For example, in Puerto Rico, year-round cultivation of Bt (Cry1Fa) maize (event TC1507) led to a rapid development of 1000-fold resistance in *S. frugiperda*, forcing premature withdrawal of the event from the market [19]. Although no resistance to Bt toxins has been reported yet in *S. exigua*, the increasing prominence of resistance issues underscores the importance of implementing insect resistance management (IRM) strategies.

Currently, the “high-dose/refuge” strategy is globally recognized as the primary approach to delaying the evolution of Bt resistance, i.e., planting Bt crops that express toxin levels significantly exceeding (typically 25-fold) the amount required to kill susceptible larvae, in combination with planting adjacent or nearby non-Bt refuge crops to sustain susceptible pest populations [10,20,21]. Refuges slow the rise in resistance allele frequency by sustaining populations of susceptible individuals that can interbreed with resistant individuals emerging from Bt plants, producing heterozygous progeny that are typically eliminated by Bt toxins within the same or subsequent growing season [21,22]. However, under refuge scenarios—particularly the seed-mixture refuge pattern—Bt and non-Bt plants are closely interspersed, enabling larvae to move short distances and select between plant types. Such movement may reduce larval exposure to effective Bt toxin doses, resulting in a low-dose exposure scenario that favors the survival of resistant heterozygotes while reducing that of susceptible individuals, ultimately weakening control efficacy and accelerating resistance evolution [23]. Furthermore, if target pests can detect and avoid Bt plants or tissues, they may be exposed to sublethal concentrations of Bt toxins, further enhancing resistance survival and hastening resistance development [24,25]. Although most studies indicate that lepidopteran females generally exhibit no oviposition preference between Bt and non-Bt crops [26,27,28], those assessments often overlook the influence of herbivore-induced plant damage. Females of many species can detect herbivore-induced plant volatiles emitted from conspecific-infested plants and subsequently avoid oviposition, as seen in *Chilo suppressalis* (Walker) and *S. frugiperda* [28,29]. In field conditions, Bt crops typically sustain less damage than non-Bt crops due to Bt toxin protection. Consequently, if females preferentially oviposit on Bt rather than refuge plants, such non-random oviposition behavior may increase the proportion of offspring developing under Bt pressure, accelerating resistance evolution and undermining the refuge strategy’s effectiveness in resistance management [30].

Previous research has examined the effects of Bt crops on oviposition, feeding, and dispersal behaviors of key pests such as *S. frugiperda*, *Helicoverpa armigera* (Hübner), and *Busseola fusca* (Fuller), as well as the impacts of Bt-containing diets and plants on the growth and development of *S. exigua* [16,28,30,31,32,33,34]. However, knowledge remains limited on how Bt maize influences larval feeding and dispersal behaviors and adult oviposition in *S. exigua*. Therefore, this study evaluated the larval feeding and oviposition preferences of *S. exigua* on Bt and non-Bt maize under laboratory conditions, and complemented this with field trials examining adult oviposition preferences and larval dispersal characteristics under a seed-mixture refuge pattern. The results aim to provide empirical evidence and insights for pest management and resistance mitigation strategies.

## 2. Materials and Methods

### 2.1. Insects and Maize Materials

The *S. exigua* population used in the experiment was collected in April 2023 from a fresh maize field in Jiangcheng County, Yunnan Province, China (22°38′12.29″ N, 101°52′19.01″ E), and maintained as a laboratory colony. Larvae were individually reared on an artificial diet consisting of wheat bran and soybean powder until pupation [35]. Newly emerged adults (♀:♂ = 10:10) were maintained in plastic containers (diameter: 12 cm; height: 9 cm) covered with sterile gauze and provided with a 10% honey solution to supplement nutrition and moisture. Egg-laden gauze was collected daily and transferred into 6 cm × 8 cm zip-lock bags for hatching. Both larvae and adults were reared under controlled conditions at 26 ± 1 °C, 60% ± 10% (RH), and a photoperiod of 16 L/8 D.

The transgenic Bt maize event DBN3601T, which expresses Cry1Ab, Vip3Aa19 insecticidal proteins, and its recipient hybrid non-Bt maize “Huaxingdan 88” were both provided by Beijing DaBeiNong Biotechnology Co., Ltd., Beijing, China. Maize seeds were sown in plastic pots (diameter: 9 cm; height: 10 cm) filled with a soil–substrate mixture, with three seeds per pot. After emergence, two vigorous plants were retained in each pot. All maize plants were cultivated in a greenhouse at 28 ± 1 °C, 80% ± 5% (RH), under a natural photoperiod.

### 2.2. Laboratory Choice Test on Feeding Preference of Spodoptera exigua Larvae to Bt and Non-Bt Maize Leaves

A leaf disc assay was conducted following the method described by Tang et al. [36], to assess the feeding preference of *S. exigua* larvae for Bt versus non-Bt maize leaves. When greenhouse-grown maize plants reached the V5–V6 growth stage, the newest fully expanded leaves were collected and punched into 2.5 cm diameter discs using a circular leaf puncher. These discs were subsequently used in two experimental setups:(a)Continuous feeding on single-type maize leaves: Each assay was conducted in a 20 cm diameter Petri dish lined with moistened filter paper divided into eight identical sectors. Eight leaf discs of the same maize type (Bt or non-Bt), each 2.5 cm in diameter, were evenly spaced along the dish perimeter (for 5th instar larvae, each position contained two overlapping discs, totaling 16 discs). After a 6 h starvation period, larvae from 1st to 5th instar were released into the center of the dish (20 larvae for 1st–3rd instars; 5 larvae for 4th–5th instars). The dish was covered with a double-layered black cotton cloth to ensure darkness and then placed in a climate chamber at 26 ± 1 °C, 60% ± 10% RH, with a photoperiod of 16 L/8 D. Larval positions and survival were recorded at 1, 2, 4, 6, 8, 10, 12, 24, 36, and 48 h post-release. The percentage of larvae observed feeding on maize leaves at each time point was calculated to evaluate feeding preference. Each treatment was replicated eight times. After 48 h, the leaf consumption area was measured using a transparent 1 mm × 1 mm grid film.(b)Continuous feeding choice test with alternating Bt and non-Bt leaves: A 20 cm diameter Petri dish was lined with moistened filter paper divided into eight equal sectors. Eight leaf discs (2.5 cm in diameter), alternating between Bt and non-Bt maize, were placed at equal intervals along the perimeter (for 5th instar larvae, each position contained two overlapping discs, totaling 16 discs). The outer edge of each sector was marked to indicate whether the leaf was from Bt or non-Bt maize. Larvae were released at the center of the dish, and rearing conditions and observation time points were identical to those described in experiment (a). Each treatment was replicated eight times. After 48 h, the area of leaf tissue consumed from Bt and non-Bt maize was measured separately.

### 2.3. Laboratory Assay of Oviposition Preference of Spodoptera exigua Adults for Bt and Non-Bt Maize Plants

The following plant treatments were prepared: (a) healthy Bt maize; (b) healthy non-Bt maize; (c) Bt maize infested with *S. exigua* larvae; and (d) non-Bt maize infested with *S. exigua* larvae. All plants were obtained from the aforementioned greenhouse cultivation and were at the V4 growth stage, with two seedlings per pot. For the larval infestation treatment, eight neonates were manually placed into the whorls of each Bt or non-Bt maize plant (treatments c and d) and allowed to feed continuously for 48 h. Larvae were retained on the plants throughout the exposure period to maintain feeding damage.

Adult oviposition preference was evaluated under six experimental treatments: (i) a only; (ii) b only; (iii) a vs. b; (iv) a vs. d; (v) b vs. c; and (vi) c vs. d. In each treatment, ten pairs of newly emerged *S. exigua* adults (♀:♂ = 1:1) were released into a rearing cage (60 cm × 60 cm × 60 cm, mesh size 120) placed in a greenhouse. A plastic Petri dish containing a cotton ball soaked in a 10% honey solution was placed in the center of the cage to provide supplemental nutrition and moisture. After 24 h of mating, six pots of maize plants (three Bt and three non-Bt maize) were arranged alternately and equidistantly around the inner perimeter of each cage. The number of egg masses and the total number of eggs laid on each maize type were recorded daily for five consecutive days. On the fifth day, plant damage was assessed based on the criteria described by Williams et al. [37]. Each treatment was replicated four times. The experiment was conducted in a greenhouse under controlled conditions of 26 ± 1 °C, 60% ± 10% RH, with a photoperiod of 16 L:8 D.

### 2.4. Feeding, Dispersal, and Oviposition Behavior of Spodoptera exigua in Bt Maize Fields Under Different Seed-Mixture Refuge Patterns

A field cage experiment was conducted from 16 September to 13 October 2024, at the Jiangcheng Experimental Station (22°31′15.57″ N, 101°29′44.54″ E) of the Institute of Plant Protection, Chinese Academy of Agricultural Sciences, using a completely randomized block design. The release of laboratory-reared insects enabled a precise assessment of insect behavior, minimizing the confounding effects of environmental variability, such as extreme rainfall or strong wind events characteristic of spring and summer. Five treatments were established to assess the feeding, dispersal, and oviposition behavior of *S. exigua* in maize fields with different seed-mixture refuges: seed-mixture refuges at 5%, 10%, and 20% proportions, pure Bt maize fields (0% refuge), and pure non-Bt maize fields (100% refuge). Each plot measured 25 m^2^ (5 m × 5 m), with a plant spacing of 30 cm and row spacing of 60 cm. To ensure germination, two seeds were sown per hole, and base fertilizer (N:P:K = 14:5:7) was applied at a rate of approximately 300 kg/ha. For the 5%, 10%, and 20% seed-mixture refuge treatments, the positions of Bt and non-Bt maize plants were randomly generated using Microsoft Excel 2021 (Microsoft Corp., Redmond, WA, USA). Non-Bt maize plants were marked with labeled tags. After sowing, all plots were fully enclosed with 80-mesh nylon netting to prevent interference from external insects. At the V2 stage, the weaker seedling in each hole was removed, leaving only one healthy plant per position.

At the V3–V4 growth stage of maize, 50 pairs of newly emerged *S. exigua* adults were released into each plot in the evening. Following adult release, the number of egg masses and total eggs laid on Bt and non-Bt maize plants was recorded daily for eight consecutive days, beginning on the first day. The coordinate location of each plant bearing egg masses was also recorded to evaluate oviposition preference. To examine larval feeding, dispersal behavior, and consequent plant damage during the 1st–4th instars under different seed-mixture refuge patterns, larval survival was monitored daily for 11 days following hatching (i.e., day 4 after adult release, designated as day 1 of larval dispersal) on Bt and non-Bt maize. During this period, the number of larvae and the coordinate location of each plant where they were observed were recorded. Based on these data, the maximum and mean dispersal distances of larvae were calculated to analyze dispersal patterns under different refuge treatments. On day 14 post-adult release, the damage level of Bt and non-Bt maize in each plot was assessed by recording the percentage of damaged plants, which served as an indicator of larval feeding behavior.

### 2.5. Statistical Analysis

The feeding selection rate was calculated using the following formula:Feeding selection rate = (Number of larvae selecting Bt or non-Bt leaf disc)/(Total number of larvae tested) × 100%(1)

All statistical analyses were performed in SPSS 26.0 (IBM, Armonk, NY, USA). Percentage data were arcsine square-root-transformed to meet normality assumptions before analysis. Independent samples *t*-tests compared the total number of egg masses and eggs laid on Bt and non-Bt maize in laboratory oviposition assays. Differences in leaf area consumed and larval feeding selection rates between Bt and non-Bt maize in laboratory assays, as well as the total number of egg masses and eggs in field cage experiments, were analyzed using the Mann–Whitney U test. In the laboratory experiment, differences in larval feeding selection rate between Bt and non-Bt maize, as well as the mean and maximum dispersal distances of larvae on maize plants in the field, were analyzed using two-way ANOVA. On day 11 after larval hatching, the mean and maximum dispersal distances of larvae and the percentage of damaged plants in the field were analyzed via one-way ANOVA. To describe the temporal dynamics of larval feeding preference and dispersal, several nonlinear regression models (quadratic and cubic polynomial functions) were fitted to the data. Model performance was evaluated using *F*- and *p*-values, the coefficient of determination (*R*^2^), and the Akaike Information Criterion (AIC), and biological plausibility was also considered. Based on these criteria, cubic polynomial functions were selected for feeding preference, whereas quadratic functions were retained for larval dispersal. Detailed model comparison results and the rationale for final model selection are provided in Appendix A.

## 3. Results

### 3.1. Laboratory Choice Test on Feeding Preference of Spodoptera exigua Larvae to Bt and Non-Bt Maize Leaves

During the 48 h continuous feeding period, the proportion of *S. exigua* larvae feeding on Bt maize leaves differed significantly from that on non-Bt maize for the 1st, 2nd, 3rd, and 5th instars (1st instar: *F*_1,140_ = 36.629, *p* = 0.000; 2nd instar: *F*_1,140_ = 283.821, *p* = 0.000; 3rd instar: *F*_1,140_ = 73.673, *p* = 0.000; 5th instar: *F*_1,140_ = 13.558, *p* = 0.000), while no significant difference was observed for the 4th instar (*F*_1,140_ = 0.751, *p* = 0.387). Over time, the feeding selection rates of 1st to 5th instar larvae gradually decreased on Bt maize while increasing on non-Bt maize (Figure 1A–E). After 48 h, the feeding selection rates of 1st, 2nd, 3rd, and 4th instar larvae on Bt maize were 9.34%, 30.63%, 52.26%, and 76.88%, respectively, all significantly lower than those on non-Bt maize (67.88%, 76.88%, 86.88%, and 96.88%, respectively) (1st instar: Z = −3.386, *p* = 0.001; 2nd instar: Z = −2.539, *p* = 0.011; 3rd instar: Z = −2.731, *p* = 0.006; 4th instar: Z = −2.385, *p* = 0.017) (Figure 1F). For 5th instar larvae, no significant difference in feeding selection rates was observed between Bt maize (90.00%) and non-Bt maize (100.00%) (Z = −1.461, *p* = 0.144) (Figure 1F). Correspondingly, leaf area consumed by larvae of all instars was significantly lower on Bt maize than on non-Bt maize (1st instar: Z = −3.291, *p* = 0.001; 2nd instar: Z = −3.381, *p* = 0.001; 3rd instar: Z = −3.406, *p* = 0.001; 4th instar: Z = −3.361, *p* = 0.001; 5th instar: Z = −3.361, *p* = 0.001) (Figure 1G). Notably, 3rd instar larvae consumed 166.92-fold more leaf area on non-Bt maize than on Bt maize, while the difference for 5th instar larvae was 13.71-fold.

In choice tests with alternating Bt and non-Bt maize leaves, 1st to 5th instar *S. exigua* larvae showed significantly lower feeding selection rates on Bt maize compared to non-Bt maize (1st instar: *F*_1,140_ = 35.374, *p* = 0.000; 2nd instar: *F*_1,140_ = 153.775, *p* = 0.000; 3rd instar: *F*_1,140_ = 44.093, *p* = 0.000; 4th instar: *F*_1,140_ = 46.197, *p* = 0.000; 5th instar: *F*_1,140_ = 5.008, *p* = 0.027). Over time, the feeding selection rates on Bt maize gradually decreased for the 1st (Figure 2A), 2nd (Figure 2B), and 5th instars (Figure 2E), while those on non-Bt maize increased. The 3rd instar larvae exhibited a fluctuating pattern on Bt maize, with selection rates first decreasing, then increasing, and decreasing again, while showing a slow increase on non-Bt maize (Figure 2C). The feeding selection rates of 4th instar larvae decreased slowly on Bt maize and initially decreased, then increased, on non-Bt maize (Figure 2D). After 48 h, the selection rates on Bt maize for 1st, 2nd, 3rd, and 4th instar were 9.38%, 18.75%, 22.50%, and 32.50%, respectively, all significantly lower than those on non-Bt maize (48.32%, 66.25%, 40.63%, and 50.00%, respectively) (1st instar: Z = −3.383, *p* = 0.001; 2nd instar: Z = −3.381, *p* = 0.001; 3rd instar: Z = −2.127, *p* = 0.033; 4th instar: Z = −2.365, *p* = 0.018) (Figure 2F). The 5th instar exhibited no significant difference between Bt (33.75%) and non-Bt maize (56.25%) (Z = −1.882, *p* = 0.06) (Figure 2F). Correspondingly, the feeding area on Bt maize leaves was significantly lower than that on non-Bt maize for all instars (1st instar: Z = −3.414, *p* = 0.001; 2nd instar: Z = −3.366, *p* = 0.001; 3rd instar: Z = −3.381, *p* = 0.001; 4th instar: Z = −3.361, *p* = 0.001; 5th instar: Z = −3.256, *p* = 0.001) (Figure 2G).

### 3.2. Laboratory Assay of Oviposition Preference of Spodoptera exigua Adults on Bt and Non-Bt Maize Plants

When confined to either Bt or non-Bt maize plants, *S. exigua* adults laid an average of 7.25 egg masses and 430.0 eggs on Bt maize, and 7.50 egg masses and 343.5 eggs on non-Bt maize, with no significant differences between plant types (egg masses: *t* = −1.20, *df* = 6, *p* = 0.908; eggs: *t* = 0.913, *df* = 6, *p* = 0.397) (Figure 3A,B). Similarly, when healthy Bt and non-Bt plants were available simultaneously, oviposition did not differ significantly between the two (egg masses: *t* = −1.414, *df* = 6, *p* = 0.207; eggs: *t* = 1.508, *df* = 6, *p* = 0.182) (Figure 3D,E).

Considering the differences in damage levels between Bt and non-Bt maize plants in the field, neonates were used to artificially damage the plants. Bt maize plants suffered significantly less damage than non-Bt maize plants (*t* = 33.295, *df* = 94, *p* = 0.000) (Figure 3C). When Bt and non-Bt maize plants with varying levels of damage coexisted, *S. exigua* adults laid more eggs on healthy Bt plants (73.55% of total oviposition) or slightly damaged Bt plants (66.67%) (Figure 3D,E). Oviposition differed significantly between healthy Bt and damaged non-Bt plants (egg masses: *t* = 5.782, *df* = 6, *p* = 0.001; eggs: *t* = 6.075, *df* = 6, *p* = 0.001), and between damaged Bt and damaged non-Bt plants (egg masses: *t* = 15.558, *df* = 6, *p* = 0.001; eggs: *t* = 35.426, *df* = 6, *p* = 0.000). Total egg deposition on healthy and damaged Bt maize plants was 2.78- and 2.00-fold higher, respectively, than that on damaged non-Bt maize plants. No significant difference was observed between damaged Bt and healthy non-Bt plants (egg masses: *t* = −0.075, *df* = 6, *p* = 0.943; eggs: *t* = −1.237, *df* = 6, *p* = 0.262).

### 3.3. Feeding, Dispersal, and Oviposition Behavior of Spodoptera exigua in Bt Maize Fields Under Different Seed-Mixture Refuge Patterns

During the field trial at the V4–V6 growth stages, *S. exigua* moths predominantly laid eggs on basal stems and abaxial (underside) surfaces of leaves. Over eight consecutive days, no significant differences were observed in the total number of egg masses or eggs among the different seed-mixture refuge treatments of 0% (100% Bt), 5%, 10%, 20%, and 100% (0% Bt) (egg masses: *F*_4,10_ = 0.865, *p* = 0.518; egg numbers: *F*_4,10_ = 0.207, *p* = 0.929) (Figure 4A–C). Specifically, the average number of egg masses under the 0%, 5%, 10%, 20%, and 100% refuge treatments was 16.33, 12.00, 11.33, 15.33, and 18.33, respectively, with corresponding egg numbers of 663.33, 562.67, 594.33, 645.00, and 783.00. Under 5%, 10%, and 20% seed-mixture refuges, oviposition on Bt maize averaged 4.89, 4.78, and 6.94 egg masses per 50 plants, with 229.24, 256.17, and 298.26 eggs, respectively; on non-Bt maize, egg masses averaged 5.56, 4.17, and 4.17 per 50 plants, with 333.33, 170.83, and 150.69 eggs. No significant differences were observed between Bt and non-Bt maize plants for either egg mass number (5% refuge: Z = −0.664, *p* = 0.507; 10% refuge: Z = −0.218, *p* = 0.827; 20% refuge: Z = −1.771, *p* = 0.077) or egg number (5% refuge: Z = −0.655, *p* = 0.513; 10% refuge: Z = −0.218, *p* = 0.827; 20% refuge: Z = −1.964, *p* = 0.050) (Figure 4D,E).

Under the seed-mixture refuge patterns, the proportion of non-Bt maize significantly influenced *S. exigua* larval dispersal distances (mean: *F*_4,165_ = 89.450, *p* = 0.000; maximum: *F*_4,165_ = 70.695, *p* = 0.000) (Figure 5A,B) (Table 1), with both the mean and maximum larval dispersal distances increased significantly with the rising proportion of non-Bt maize. On day 11, the mean dispersal distance in the 100% refuge reached 30.32 cm, which was significantly greater than that in the 0% refuge at 0.00 cm. The mean dispersal distances under the 20%, 10%, and 5% refuge patterns were 19.91 cm, 3.33 cm, and 0.00 cm, respectively, showing a decreasing trend (*F*_4,10_ = 50.711, *p* = 0.000) (Figure 5C). Maximum dispersal in the 100% refuge was 84.85 cm, also significantly higher than that in the 0% refuge at 0.00 cm. Under the 20%, 10%, and 5% refuge models, the maximum dispersal distances were 64.72 cm, 10.00 cm, and 0.00 cm, respectively (*F*_4,10_ = 75.731, *p* = 0.000) (Figure 5D). Survival larvae in the field were predominantly at the 2nd or 3rd instar, with only a few individuals reaching the 4th instar (Figure 5E). No significant differences in plant damage rate were detected among non-Bt or Bt maize plants across different refuge proportions (non-Bt maize: *F*_3,8_ = 0.033, *p* = 0.991; Bt maize: *F*_3,8_ = 1.000, *p* = 0.441) (Figure 5F). Bt maize plants remained entirely undamaged under the 5% and 10% refuge conditions, whereas in the 20% refuge, only one Bt maize plant showed minor feeding damage.

## 4. Discussion

Our previous research demonstrated that *S. exigua* larvae are highly susceptible to Bt toxins expressed in DBN3601T maize [16]. This high susceptibility drives larvae to preferentially feed on refuge non-Bt maize plants while exhibiting pronounced antifeedant and avoidance behaviors toward Bt maize. Under field-based refuge planting scenarios, Bt maize effectively restricted the long-distance dispersal of larvae. In contrast, the oviposition behavior of *S. exigua* females was not significantly influenced by the presence of Bt toxins, as females could not distinguish between healthy Bt and non-Bt maize plants. However, when non-Bt maize plants were severely damaged, females tended to oviposit on undamaged or slightly damaged Bt maize plants.

When herbivorous insects concentrate feeding on particular plant tissues, this typically reflects either a preference for those tissues or avoidance of others. This study showed that *S. exigua* larvae from the 1st to 5th instars exhibited antifeedant and avoidance behaviors toward Bt maize leaves, with these behaviors more pronounced in younger instars. This observation is consistent with most previous findings [32,38,39,40]. For example, under choice conditions, the feeding behavior of resistant populations of *Mythimna unipuncta* (Haworth) was less affected by MON810 maize leaf tissue than that of susceptible populations [41]. Additionally, younger instars of *Spodoptera litura* (Fabricius) and *H. armigera* exhibited significantly reduced crawling and dispersal abilities after feeding on Bt cotton or Bt-containing diets, whereas older instars were less affected. In some cases, older instars even exhibited increased short-distance movement or escape behaviors in response to certain doses of Bt toxins [42]. This behavior likely reflects post-ingestive adverse effects of Bt toxins. Even brief feeding can deliver Bt toxins to bind receptors on the midgut brush-border membrane, rapidly compromising gut integrity and triggering acute physiological stress [43,44]. Neural feedback from this disturbance may induce a conditioned aversion, leading larvae to cease feeding and disperse. In parallel, gustatory mechanisms may also contribute. Insects use gustatory sensilla to assess host acceptability through probing or test biting and to detect plant metabolites [45,46]. Whether *S. exigua* larvae can perceive Bt toxins—or Bt-plant–associated secondary metabolites—via gustatory sensilla and thereby trigger rapid taste aversion at the onset of feeding remains to be determined. Field trials further confirmed that larval dispersal was inhibited in Bt maize fields, with the maximum dispersal distance reaching only 50.00 cm, compared to 84.85 cm in non-Bt maize fields. However, the frequency and likelihood of short-distance movement between plants might increase. This behavior likely stems from the larvae’s inherent tendency to avoid Bt toxins, which aligns with our previous findings that *S. exigua* larvae cannot survive on DBN3601T maize [16]. Similarly, Goldstein et al. found that *O. nubilalis* larvae exhibited a higher frequency of silk-threaded escape behavior on Bt maize than on non-Bt maize [47]. Prior work across Lepidoptera (e.g., *S. frugiperda*, *B. fusca*, *Helicoverpa zea* (Boddie)) shows that early instar larvae are more active and mobile, whereas later instars exhibit reduced mobility and greater concealment [48,49,50]. Future studies should monitor larvae through pupation and across multiple generations to assess how later-instar behavior influences overall damage and resistance management. Overall, larval susceptibility to Bt toxins significantly influenced feeding and dispersal behavior. Such behavioral adjustments may simultaneously reduce feeding on Bt maize—potentially increasing yield—while decreasing toxin intake, thereby compromising the pest-control efficacy of Bt maize.

Typically, adult herbivorous insects preferentially oviposit on host plants that provide optimal nutrition and conditions suitable for larval development, thereby enhancing offspring survival [51]. Theoretically, if Bt crops impose significant fitness costs on progeny, female moths should evolve strong oviposition preferences to avoid these plants. However, our study indicated that *S. exigua* females were unable to distinguish between healthy Bt and non-Bt maize plants, consistent with several previous studies examining pest oviposition preferences on Bt crops [28,31,52,53]. This suggests that *S. exigua* females cannot detect or recognize the Bt toxins expressed by Bt crop plants. In nature, female moths can distinguish between damaged and undamaged hosts through plant-emitted volatiles and detect conspecific egg-associated cues, thereby preferentially ovipositing on undamaged plants [28,54,55]. For instance, rice volatiles such as 2-heptanol, α-cedrene, and β-myrcene emitted following damage by *C. suppressalis* larvae repel female moths, leading them to prefer oviposition on undamaged Bt or non-Bt rice plants [28]. Our laboratory oviposition assays showed that *S. exigua* adults tended to lay eggs on undamaged or lightly damaged Bt maize plants, consistent with oviposition preferences observed in other lepidopterans such as *C. suppressalis* and *S. frugiperda* [28,30,56]. Such behavior may influence pest population dynamics and has important implications for pest management. For instance, Bt maize could potentially serve as a “dead-end trap crop” in the field, attracting oviposition while eliminating larvae through its insecticidal activity, thereby diverting them from non-Bt maize and reducing pest pressure on the latter [28,57]. However, in field trials, oviposition did not differ between Bt and non-Bt maize despite conspecific feeding injury on non-Bt plants. This likely reflects the low damage levels in our plots (only 9.72% of non-Bt plants damaged 14 days after release). Similarly, a three-year field survey by Zhao et al. found that light damage to non-Bt maize did not induce a strong shift in *S. frugiperda* oviposition toward DBN3601T maize [58]. Moreover, under field conditions, wind and other meteorological factors might disrupt the directional dispersion of plant volatiles, and the complex “olfactory background” generated by volatiles from all surrounding plants may further reduce the ability of female moths to accurately locate specific host plants [59,60]. Changes in maize plant volatiles caused by larval feeding may explain the differences in attraction for *S. exigua* oviposition. However, the specific volatile compounds and their concentrations responsible for the repellency or attraction effects on female moths require further analysis and validation.

Another important implication of our study concerns the evolution of Bt resistance in target pests, which poses a major threat to the sustainable use of Bt crops. Currently, the primary strategy to delay Bt resistance evolution is the deployment of non-Bt refuges; however, the optimal spatial configuration of these refuges remains debated. Some mathematical models suggest that when larvae frequently move between plants, seed-mixture refuges may accelerate the accumulation of resistance alleles compared to structured refuges, as sensitive alleles are harder to maintain and the risk of resistance allele dominance is increased under mixed planting [21,61,62,63]. From an agronomic perspective, however, seed-mixture refuges are easier to implement, can significantly reduce farmer non-compliance, and lower implementation costs [21]. Our field trials were conducted during autumn maize growth stage, when temperatures, monsoon activity, and rainfall were relatively low. Under these conditions, the maximum dispersal distance of *S. exigua* larvae on non-Bt refuge maize was 84.85 cm, which exceeded that reported for *S. frugiperda* under indoor still-air conditions (71 cm), but was lower than the distance observed under windy conditions (196 cm) [48]. Larval dispersal is influenced by various abiotic factors, such as temperature, wind, and rainfall, as well as biotic factors [64,65,66]. The comparatively constrained dispersal we observed in *S. exigua* likely reflects the specific climatic conditions during the autumn trials, consistent with Kang et al. [67], who reported reduced *S. frugperda* movement under winter conditions relative to summer. Our results therefore reflect larval behavior under specific seasonal and developmental windows, and extrapolation to other cropping seasons or agroecological contexts should be approached with caution. Future multi-season field studies are needed to more comprehensively evaluate larval dispersal behavior, as well as the long-term stability and broader applicability of seed-mixture refuge strategies. Further investigation showed that increasing the proportion of refuge non-Bt maize increased larval dispersal distance: larvae dispersed farther at 20% refuge than at 5% or 10%. Bt maize sustained no significant injury at 5% and 10% refuge, whereas individual Bt plants showed feeding damage at 20%. These results indicate that mixed refuges may be a suitable strategy for managing *S. exigua* populations, while also highlighting the potential risks associated with larval dispersal between plants. From a behavioral-ecology perspective, the dispersal patterns of *S. exigua* larvae are not only constrained by environmental factors but may also reflect active foraging strategies under conditions of limited or unevenly distributed resources. At low refuge levels (5% and 10%), larvae encountered Bt toxin more rapidly and frequently, resulting higher mortality and reduced local population density. This situation may suppress locomotion and diminish incentives to depart the natal plant. At higher refuge (20%), the greater availability of non-Bt hosts can sustain higher local densities and accelerate resource depletion, thereby promoting a higher propensity for interplant movement. By day 11, larval density at the 20% refuge was marginally higher than at the 10% and 5% refuges. Such resource-driven dispersal and feeding behavior may accelerate the evolution of resistance by reducing effective Bt toxin exposure, lowering survival of susceptible individuals, or increasing the relative survival of heterozygotes [21,63]. Our study provides critical data on the behavioral responses of *S. exigua* larvae exposed to Bt toxins, enhancing understanding of larval dispersal behavior and its key role in host selection, especially in the context of seed-mixture refuges where Bt and non-Bt maize plants are adjacent.

Currently, refuge strategies to delay pest resistance to Bt crops are based on the assumption that target pest adults mate and oviposit randomly on Bt and non-Bt plants [22,68]. Our study found that when non-Bt maize plants were damaged, *S. exigua* females tend to avoid ovipositing on these plants. This non-random oviposition behavior may accelerate the evolution of Bt resistance by increasing the proportion of the population under selective pressure, thereby weakening the effectiveness of refuge strategies in resistance management. Although no oviposition preference was observed between Bt and non-Bt maize plants under refuge field conditions, with increasing damage rates on non-Bt maize, adults gradually laid more eggs on Bt maize plants than on non-Bt maize plants. Model analyses by Téllez-Rodríguez et al. indicated that a strong oviposition preference of *S. frugiperda* for Bt maize could significantly accelerate the evolution of resistance to Bt traits, potentially necessitating larger refuges or even undermining the long-term efficacy of resistance management strategies [30]. Therefore, when designing Bt maize resistance management strategies in China—especially in regions south of the Tropic of Cancer where *S. exigua* can reproduce year-round—this factor should be fully considered. During severe outbreaks of *S. exigua*, it is recommended to apply chemical insecticides to effectively control pest damage on maize plants, thus avoiding or reducing the accumulation of resistance alleles.

This study preliminarily revealed the behavioral effects of Bt maize on *S. exigua* through laboratory assays and short-term field experiments. However, comprehensive evaluation of the efficacy of Bt crops under natural conditions and the risk of resistance evolution requires further long-term field monitoring. In particular, assessments of oviposition and mating behaviors, as well as larval damage patterns on Bt and non-Bt maize, are essential. Such efforts will contribute to a deeper understanding of pest behavior under actual agricultural conditions.

## 5. Conclusions

With their high efficacy in controlling major agricultural pests, Bt crops have rapidly expanded in cultivation worldwide, reshaping crop production and protection practices in many regions [69]. Through both laboratory and field experiments, this study demonstrates that *S*. *exigua* larvae exhibit pronounced antifeedant and avoidance behaviors toward Bt maize, and that Bt maize under field conditions effectively inhibits larval dispersal. *S. exigua* female moths preferentially oviposit on undamaged or slightly damaged Bt plants when non-Bt maize plants are heavily injured. These findings provide empirical support for optimizing the effectiveness of Bt crops in pest management and refuge design for insect resistance management.

## Figures and Tables

**Figure 1 insects-16-01059-f001:**
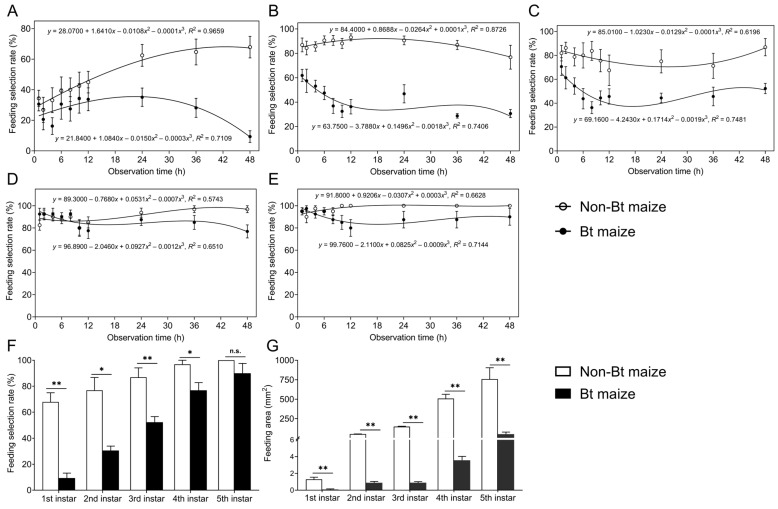
Changes in feeding selection rates of 1st (**A**), 2nd (**B**), 3rd (**C**), 4th (**D**), and 5th (**E**) instar *Spodoptera exigua* larvae on Bt or non-Bt maize leaf tissues under continuous feeding on single-type maize leaves; (**F**) feeding selection rates of 1st–5th instar larvae on Bt or non-Bt maize leaves at 48 h; (**G**) feeding area of 1st–5th instar larvae on Bt or non-Bt maize leaves. Asterisks (*) indicate significant differences: * *p* < 0.05; ** *p* < 0.01; n.s. indicates no significant difference (*p* > 0.05) (Mann–Whitney U-test).

**Figure 2 insects-16-01059-f002:**
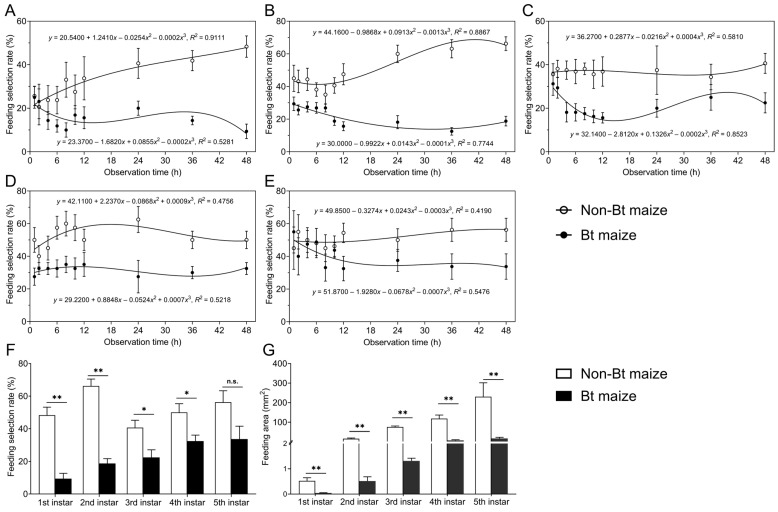
Changes in feeding selection rates of 1st (**A**), 2nd (**B**), 3rd (**C**), 4th (**D**), and 5th (**E**) instar *Spodoptera exigua* larvae in the choice test with alternating Bt and non-Bt maize leaves; (**F**) feeding selection rates of 1st–5th instar larvae on Bt or non-Bt maize leaves at 48 h; (**G**) feeding area of 1st–5th instar larvae on Bt or non-Bt maize leaves. Asterisks (*) indicate significant differences: * *p* < 0.05; ** *p* < 0.01; n.s. indicates no significant difference (*p* > 0.05) (Mann–Whitney U-test).

**Figure 3 insects-16-01059-f003:**
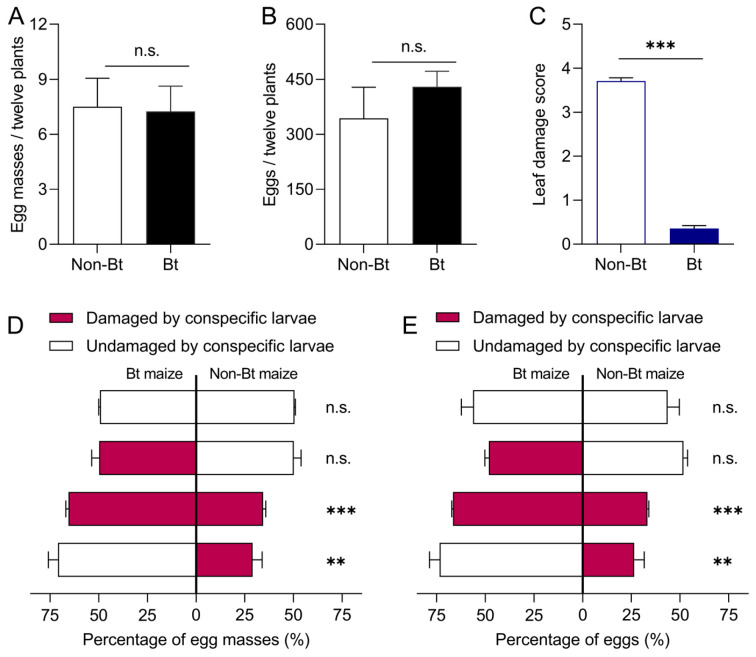
Oviposition preference of *Spodoptera exigua* adults on Bt and non-Bt maize plants. (**A**) Total number of egg masses on Bt and non-Bt maize plants under non-choice conditions; (**B**) corresponding total number of eggs. (**C**) Damage caused by larvae to Bt and non-Bt maize plants. (**D**,**E**) Proportion of egg masses and eggs laid on damaged and undamaged Bt and non-Bt maize plants under choice conditions. Asterisks (*) indicate significant differences: ** *p* < 0.01; *** *p* < 0.001; n.s. indicates no significant difference (*p* > 0.05) (Independent sample *t*-test).

**Figure 4 insects-16-01059-f004:**
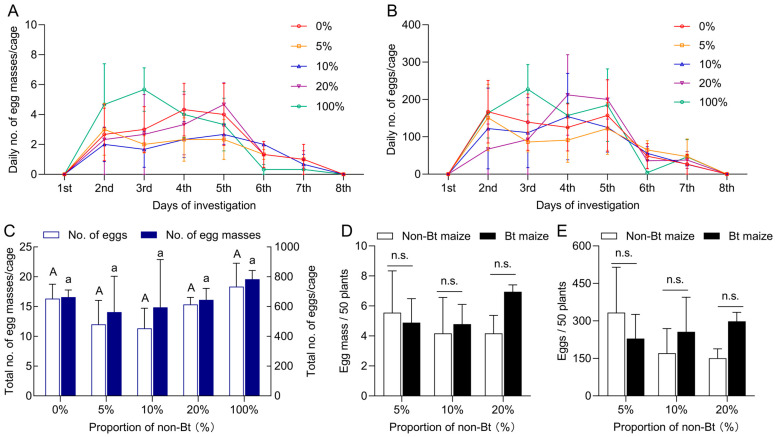
Oviposition behavior of *Spodoptera exigua* adults under different seed-mixture refuge patterns. (**A**) Daily number of egg masses; (**B**) daily number of eggs; (**C**) total number of egg masses and eggs under 0%, 5%, 10%, 20%, and 100% seed-mixture refuge patterns. Distribution of egg masses (**D**) and eggs (**E**) on Bt and non-Bt maize plants under the 5%, 10%, and 20% seed-mixture refuge patterns. Different lowercase and uppercase letters indicate significant differences in egg masses or egg numbers among different refuges (*p* < 0.05) (One-way ANOVA, Tukey’s HSD). n.s. indicates no significant difference between Bt and non-Bt maize plants within the same refuge pattern (*p* > 0.05) (Mann–Whitney U-test).

**Figure 5 insects-16-01059-f005:**
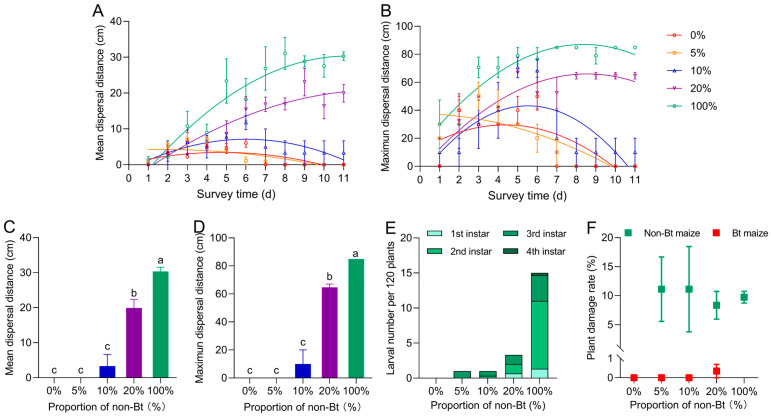
Mean (**A**) and maximum (**B**) dispersal distances of *Spodoptera exigua* larvae under different seed-mixture refuge proportions. On Day 11, larval average dispersal distance (**C**), maximum dispersal distance (**D**), larval number per 120 plants (**E**), and plant damage rates (**F**) under different seed-mixture refuge proportions. Different lowercase letters indicate significant differences in larval dispersal distances and plant damage rates among different seed-mixture refuge proportions (*p* < 0.05) (One-way ANOVA, Tukey’s HSD).

**Table 1 insects-16-01059-t001:** Quadratic equations fitted to mean and maximum dispersal distances of *Spodoptera exigua* larvae under different refuge patterns.

Treatment	Mean Dispersal Distances		Maximum Dispersal Distances	
	Fitted Equation	*R* ^2^	Fitted Equation	*R* ^2^
0% (Pure Bt maize)	*y* = 0.450 + 1.264*x* − 0.132*x*^2^	0.448	*y* = 11.210 + 8.427*x* − 0.967*x*^2^	0.454
5% Seed-mixture	*y* = 4.116 + 0.211*x* − 0.066*x*^2^	0.493	*y* = 37.560 − 0.146*x* − 0.378*x*^2^	0.704
10% Seed-mixture	*y* = −1.340 + 2.803*x* − 0.233*x*^2^	0.508	*y* = −7.415 + 18.290*x* − 1.653*x*^2^	0.469
20% Seed-mixture	*y* = −4.845 + 4.010*x* − 0.157*x*^2^	0.908	*y* = −2.583 + 16.020*x* − 0.935*x*^2^	0.812
100% (Pure non-Bt maize)	*y* = −7.799 + 6.805*x* − 0.304*x*^2^	0.918	*y* = 12.720 + 17.760*x* − 1.061*x*^2^	0.874

## Data Availability

The original contributions presented in this study are included in the article/Appendix A. Further inquiries can be directed to the corresponding author.

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
