# Peer review of "Selection Behavior of the Beet Armyworm, Spodoptera exigua (Hübner) Between Bt Maize and Conventional Maize Plants"

_insects, 2025, doi:10.3390/insects16101059_

Round 1

Reviewer 1 Report

Comments and Suggestions for Authors

This is a very well-thought out and conducted study. This manuscript was well written.

This manuscript deals with movement of Spodoptera exigua on isoline and Vip3/Cry1Ab expressing maize to determine what the appropriate refuge strategy would be. As. S. exigua can be an important leaf feeder, knowledge of larval movement within a seed blend refuge is critical for durability. Also these or similar studies have been conducted with other insect species, this would be the first report for S. exigua. The methods in this MS are adequate, intensive and very straightforward. And based on these well thought out methods, it is relatively easy to come to a valid conclusion. This manuscript is also well written. 

Author Response

The point-by-point responses to the comments and suggestions from the reviewer 1

Reviewer 1: This is a very well-thought out and conducted study. This manuscript was well written.

This manuscript deals with movement of Spodoptera exigua on isoline and Vip3/Cry1Ab expressing maize to determine what the appropriate refuge strategy would be. As. S. exigua can be an important leaf feeder, knowledge of larval movement within a seed blend refuge is critical for durability. Also these or similar studies have been conducted with other insect species, this would be the first report for S. exigua. The methods in this MS are adequate, intensive and very straightforward. And based on these well thought out methods, it is relatively easy to come to a valid conclusion. This manuscript is also well written.

Response: We are very grateful for your positive comments and thoughtful recognition of this study.

Reviewer 2 Report

Comments and Suggestions for Authors

The manuscript titled "Selection Behavior of the Beet Armyworm, Spodoptera exigua (Hübner) between Bt Maize and Conventional Maize Plants," authored by Song et al., provides a comprehensive examination of the feeding and oviposition preferences of Spodoptera exigua on both conventional and Bt maize. The authors report that larvae display a preference for conventional corn over Bt corn. Furthermore, female oviposition appears to be random between the two types, although both greenhouse and field trials indicate that plants damaged by conspecific larvae are typically avoided during site selection for oviposition. Notably, larvae on non-Bt plants demonstrate a higher capacity for movement between plants, highlighting that the in-bag refuge strategy, although more convenient and easier to implement, could also contribute to the development of resistance. Overall, the findings are well-aligned with the journal's scope, and the authors contribute significant knowledge regarding this pest and strategies for Bt resistance management. I noted only minor issues that should be addressed prior to publication.

General comment:

The manuscript is well-developed, and the results are clearly presented. The figures are organized effectively. The authors provide consistent references that align with their findings.

Line 120: Regarding the continuous feeding choice test with alternating Bt and non-Bt leaves: Have you marked the Petri dish or leaves to identify which disc is Bt and which is a non-Bt leaf disc? Please include this detail to clarify the method used. The current version is unclear.

Line 191: Regarding Feeding, Dispersal, and Oviposition Behavior of Spodoptera exigua in Bt Maize Fields Under Different Seed-Mixture Refuge Patterns

I suggest the author clarify why they selected a 14-day period to assess the experiments, which is 11 days after the initial egg masses hatched. I am pointing this out because it is known that feeding capacity increases during the late instar stages. Would that impact your damage data?

What was the larvae's developmental stage on day 11 post-hatching? Please provide this detail. Lastly, late instar Lepidoptera generally exhibit reduced mobility. During your field trials, did you observe this same pattern? You may include a sentence about it.

Author Response

The point-by-point responses to the comments and suggestions from the reviewer 2

Reviewer 2: General comment: The manuscript is well-developed, and the results are clearly presented. The figures are organized effectively. The authors provide consistent references that align with their findings.

Response: We sincerely thank this reviewer for the thorough and constructive comments.

Comment 1: Line 120: Regarding the continuous feeding choice test with alternating Bt and non-Bt leaves: Have you marked the Petri dish or leaves to identify which disc is Bt and which is a non-Bt leaf disc? Please include this detail to clarify the method used. The current version is unclear.

Response: Revised. We have added this sentence to present it clearly: “The outer edge of each sector was marked to indicate whether the leaf was from Bt or non-Bt maize.” (Lines 166167).

Comment 2: Line 191: Regarding Feeding, Dispersal, and Oviposition Behavior of Spodoptera exigua in Bt Maize Fields Under Different Seed-Mixture Refuge Patterns.

I suggest the author clarify why they selected a 14-day period to assess the experiments, which is 11 days after the initial egg masses hatched. I am pointing this out because it is known that feeding capacity increases during the late instar stages. Would that impact your damage data?

What was the larvae's developmental stage on day 11 post-hatching? Please provide this detail. Lastly, late instar Lepidoptera generally exhibit reduced mobility. During your field trials, did you observe this same pattern? You may include a sentence about it.

Response: Revised. We thank the reviewer for this valuable suggestion. The objective was to compare feeding, dispersal, and plant injury during the larval period under different seed-mixture refuge configurations. Previous studies indicate that S. exigua larvae need 15–17 days to develop from hatching to pupation (Maharjan et al., 2022). Between day 1 and day 11 post-hatching (corresponding to day 14 of the experiment), individuals generally progress through the 1st to 4th instar stages, a developmental window characterized by intensive feeding activity and an elevated propensity for dispersal. Field observations confirmed that by day 11, some larvae in the 100% (non-Bt maize) had reached the 4th instar stage, while most larvae on non‑Bt maize in the 5%, 10%, and 20% refuge scenarios reached late 3rd instar. This information has been incorporated into the revised manuscript (Fig. 5E; Lines 365367).

In addition, we observed that larval dispersal distance stabilized by days 9–11 (Fig. 5), and that subsequent late-instar larvae (4th–5th instars) exhibited relatively little variation in their feeding positions in the field; therefore, no additional data were collected for these stages. To provide context on later-instar behavior, we expanded the Discussion by adding: “Prior work across Lepidoptera (e.g., S. frugiperda, B. fusca, H. zea) shows that early instar larvae are more active and mobile, whereas later instars exhibit reduced mobility and greater concealment [48-50]. Future studies should monitor larvae through pupation and across multiple generations to assess how later‑instar behavior influences overall damage and resistance management.” (Lines 418423).

Accordingly, we also revised the Materials and Methods as follow: “To examine larval feeding, dispersal behavior, and consequent plant damage during 1st-4th instars under different seed-mixture refuge patterns, larval survival was monitored daily for 11 days following hatching (i.e., day 4 after adult release, designated as day 1 of larval dispersal) on Bt and non-Bt maize.” (Lines 215219).

Maharjan, R., Ahn, J., & Yi, H. (2022). Interactive Effects of Temperature and Plant Host on the Development Parameters of Spodoptera exigua (Hübner) (Lepidoptera: Noctuidae). Insects, 13(8), 747. https://doi.org/10.3390/insects13080747.

Reviewer 3 Report

Comments and Suggestions for Authors

This manuscript presents a systematic investigation into the behavioral responses of Spodoptera exigua to Bt maize, combining laboratory and field experiments. The study is highly relevant to insect resistance management (IRM) and provides valuable empirical data. The experimental design is comprehensive and appropriate for addressing the research objectives. The introduction excellently contextualizes the problem, and the English language is of good quality.

However, several aspects require improvement to enhance the manuscript's clarity, robustness, and impact:

Materials and Methods.

  1. Line 193“A field cage experiment was conducted from 16 September to 13 October 2024......”

The experimental design is comprehensive, covering larval feeding preference, adult oviposition choice, and larval dispersal behavior in the field, thereby validating behavioral responses at multiple levels. The field trial lasted only one month (September–October 2024). Is this sufficient to represent the behavior of S. exigua across different growth stages or seasons? It is recommended to justify the timing of the experiment and discuss potential limitations for generalizing the results.

  1. Line 232-234“...... multiple nonlinear regression models were fitted.”

The manuscript mentions fitting “nonlinear regression models” to behavioral trends but does not specify the model selection criteria (e.g., AIC, BIC) or validation methods. It is advised to include goodness-of-fit tests and residual analysis to enhance statistical robustness.

Discussion

  1. Line 384-393“This observation is consistent with most previous findings... These findings suggest that younger instars are more sensitive to Bt toxins... while older instars respond only when Bt toxin accumulation... reaches a physiological threshold.”

Is the larval antifeedant response to Bt maize related to toxin perception or post-ingestive discomfort? It would be beneficial to discuss potential sensory mechanisms (e.g., gustatory receptors, neural responses) or cite supporting studies.

  1. Line 418-427“Our laboratory oviposition assays showed that exigua adults tended to lay eggs on undamaged or lightly damaged Bt maize plants......However, in field trials, despite feeding damage by conspecific larvae on non-Bt maize, no significant difference in oviposition was observed between Bt and non-Bt maize plants.”

In the laboratory, adults preferred to lay eggs on undamaged Bt plants, whereas no significant preference was observed in the field. Could this be due to interfering factors in the field environment (e.g., wind, natural enemies, complexity of plant volatiles)? It is suggested to discuss the influence of environmental variables on behavior.

  1. Line 449-453“Further investigation indicated that as the proportion of refuge non-Bt maize increased, larval dispersal distance also increased. Compared to 5% and 10% refuge, larvae exhibited stronger dispersal ability at the 20% refuge. In addition, Bt maize plants showed no significant damage at the 5% and 10% refuge, whereas individual Bt plants exhibited larval feeding damage under the 20% refuge.”

The study found that larval dispersal distance significantly increased under the 20% refuge ratio but did not explain why dispersal was limited under the 5% and 10% refuge conditions. Could this be related to population density or resource-searching strategies? Further analysis based on larval behavioral ecology is recommended.

Author Response

The point-by-point responses to the comments and suggestions from the reviewer 3

Reviewer 3: This manuscript presents a systematic investigation into the behavioral responses of Spodoptera exigua to Bt maize, combining laboratory and field experiments. The study is highly relevant to insect resistance management (IRM) and provides valuable empirical data. The experimental design is comprehensive and appropriate for addressing the research objectives. The introduction excellently contextualizes the problem, and the English language is of good quality.

However, several aspects require improvement to enhance the manuscript's clarity, robustness, and impact:

Response: We sincerely thank this reviewer for the thorough and constructive comments.

Comment 1: Materials and Methods.

Line 193“A field cage experiment was conducted from 16 September to 13 October 2024......”

The experimental design is comprehensive, covering larval feeding preference, adult oviposition choice, and larval dispersal behavior in the field, thereby validating behavioral responses at multiple levels. The field trial lasted only one month (September–October 2024). Is this sufficient to represent the behavior of S. exigua across different growth stages or seasons? It is recommended to justify the timing of the experiment and discuss potential limitations for generalizing the results.

Response: Revised. Thank you for this constructive comment. We scheduled the experiment for September–October 2024 to coincide with the local autumn maize planting season, when weather is relatively stable compared with spring and summer, thereby reducing the likelihood that extreme rainfall or abrupt temperature shifts would disrupt behavioral measurements. Accordingly, we added the following statement to the Materials and methods section: “The release of laboratory-reared insects allowed for precise assessment of insect behavior to minimize confounding effects of environmental variability, such as extreme rainfall or strong wind event characteristic in spring and summer.” (Lines 197199).

We also discussed this issue as follow: “Our field trials were conducted during autumn maize growth stage, when temperatures, monsoon activity, and rainfall were relatively low. Under these conditions, the maximum dispersal distance of S. exigua larvae on non-Bt refuge maize was 84.85 cm, which exceeded that reported for S. frugiperda under indoor still-air conditions (71 cm), but was lower than the distance observed under windy conditions (196 cm) [48]. Larval dispersal is influenced by various abiotic factors, such as temperature, wind, and rainfall, as well as biotic factors [64-66]. The comparatively constrained dispersal we observed in S. exigua likely reflects the specific climatic conditions during the autumn trials, consistent with Kang et al. [67], who reported reduced S. frugiperda movement under winter conditions relative to summer. Our results therefore reflect larval behavior under specific seasonal and developmental windows, and extrapolation to other cropping seasons or agroecological contexts should be approached with caution. Future multi-season field studies are needed to more comprehensively evaluate larval dispersal behavior, as well as the long-term stability and broader applicability of seed-mixture refuge strategies.” (Lines 468482).

Comment 2: Line 232-234“......multiple nonlinear regression models were fitted.”

The manuscript mentions fitting “nonlinear regression models” to behavioral trends but does not specify the model selection criteria (e.g., AIC, BIC) or validation methods. It is advised to include goodness-of-fit tests and residual analysis to enhance statistical robustness.

Response: Accepted. We have compared quadratic and cubic polynomial candidates using AIC and BIC and goodness-of-fit (R², F-tests). For feeding preference, quadratic and cubic models yielded similar AIC values, but the cubic models generally achieved higher R² and constituted the majority of best-fitting cases; we therefore retained the cubic form for subsequent analyses. For larval dispersal, although the cubic model minimized AIC and maximized R², it generated biologically unrealistic negative predictions at later time points. Therefore, we selected the quadratic model as the best compromise between statistical performance and biological plausibility. These procedures are now described in the Statistical Analysis section (Lines 238–245) and detailed in Supporting Information Tables S1–S3.

Comment 3: Discussion

Line 384-393“This observation is consistent with most previous findings... These findings suggest that younger instars are more sensitive to Bt toxins... while older instars respond only when Bt toxin accumulation... reaches a physiological threshold.”

Is the larval antifeedant response to Bt maize related to toxin perception or post-ingestive discomfort? It would be beneficial to discuss potential sensory mechanisms (e.g., gustatory receptors, neural responses) or cite supporting studies.

Response: Revised. We have expanded the Discussion to address potential mechanisms as follows: “This behavior likely reflects post‑ingestive adverse effects of Bt toxins. Even brief feeding can deliver Bt toxins to bind receptors on the midgut brush-border membrane, rapidly compromising gut integrity and triggering acute physiological stress [43, 44]. Neural feedback from this disturbance may induce a conditioned aversion, leading larvae to cease feeding and disperse. In parallel, gustatory mechanisms may also contribute. Insects use gustatory sensilla to assess host acceptability through probing or test biting and to detect plant metabolites [45, 46]. Whether S. exigua larvae can perceive Bt toxins—or Bt-plant–associated secondary metabolites—via gustatory sensilla and thereby trigger rapid taste aversion at the onset of feeding remains to be determined.” (Lines 402411).

Comment 4: Line 418-427“Our laboratory oviposition assays showed tha exigua adults tended to lay eggs on undamaged or lightly damaged Bt maize plants......However, in field trials, despite feeding damage by conspecific larvae on non-Bt maize, no significant difference in oviposition was observed between Bt and non-Bt maize plants.”

In the laboratory, adults preferred to lay eggs on undamaged Bt plants, whereas no significant preference was observed in the field. Could this be due to interfering factors in the field environment (e.g., wind, natural enemies, complexity of plant volatiles)? It is suggested to discuss the influence of environmental variables on behavior.

Response: As suggested, we have expanded the Discussion as follows: “Moreover, under field conditions, wind and other meteorological factors might disrupt the directional dispersion of plant volatiles, and the complex “olfactory background” generated by volatiles from all surrounding plants may further reduce the ability of female moths to accurately locate specific host plants [59, 60].” (Lines 451455).

Comment 5: Line 449-453“Further investigation indicated that as the proportion of refuge non-Bt maize increased, larval dispersal distance also increased. Compared to 5% and 10% refuge, larvae exhibited stronger dispersal ability at the 20% refuge. In addition, Bt maize plants showed no significant damage at the 5% and 10% refuge, whereas individual Bt plants exhibited larval feeding damage under the 20% refuge.”

The study found that larval dispersal distance significantly increased under the 20% refuge ratio but did not explain why dispersal was limited under the 5% and 10% refuge conditions. Could this be related to population density or resource-searching strategies? Further analysis based on larval behavioral ecology is recommended.

Response: As suggested, we have revised these in the Discussion section as follows: “From a behavioral-ecology perspective, the dispersal patterns of S. exigua larvae are not only constrained by environmental factors but may also reflect active foraging strategies under conditions of limited or unevenly distributed resources. At low refuge levels (5% and 10%), larvae encountered Bt toxin more rapidly and frequently, resulting higher mortality and reduced local population density. This situation may suppress locomotion and diminish incentives to depart the natal plant. At higher refuge (20%), the greater availability of non-Bt hosts can sustain higher local densities and accelerate resource depletion, thereby promoting a higher propensity for interplant movement. By day 11, larval density at the 20% refuge was marginally higher than at the 10% and 5% refuges. Such resource-driven dispersal and feeding behavior may accelerate the evolution of resistance by reducing effective Bt toxin exposure, lowering survival of susceptible individuals, or increasing the relative survival of heterozygotes [21, 68].” (Lines 488499).